# The Interplay between Immune System and Microbiota in Inflammatory Bowel Disease: A Narrative Review

**DOI:** 10.3390/ijms22063076

**Published:** 2021-03-17

**Authors:** Laila Aldars-García, Alicia C. Marin, María Chaparro, Javier P. Gisbert

**Affiliations:** 1Gastroenterology Unit, Hospital Universitario de La Princesa, Instituto de Investigación Sanitaria Princesa (IIS-IP), Universidad Autónoma de Madrid (UAM), 28006 Madrid, Spain; aliciacmarin@gmail.com (A.C.M.); mariachs2005@gmail.com (M.C.); javier.p.gisbert@gmail.com (J.P.G.); 2Centro de Investigación Biomédica en Red de Enfermedades Hepáticas y Digestivas (CIBEREHD), 28006 Madrid, Spain

**Keywords:** immune system, dysbiosis, inflammatory bowel disease, Crohn’s disease, ulcerative colitis, microbiota

## Abstract

The importance of the gut microbiota in human health is currently well established. It contributes to many vital functions such as development of the host immune system, digestion and metabolism, barrier against pathogens or brain–gut communication. Microbial colonization occurs during infancy in parallel with maturation of the host immune system; therefore, an adequate cross-talk between these processes is essential to generating tolerance to gut microbiota early in life, which is crucial to prevent allergic and immune-mediated diseases. Inflammatory bowel disease (IBD) is characterized by an exacerbated immune reaction against intestinal microbiota. Changes in abundance in the gut of certain microorganisms such as bacteria, fungi, viruses, and archaea have been associated with IBD. Microbes that are commonly found in high abundance in healthy gut microbiomes, such as *F. prausnitzii* or *R. hominis*, are reduced in IBD patients. *E. coli*, which is usually present in a healthy gut in very low concentrations, is increased in the gut of IBD patients. Microbial taxa influence the immune system, hence affecting the inflammatory status of the host. This review examines the IBD microbiome profile and presents IBD as a model of dysbiosis.

## 1. Introduction

Inflammatory bowel diseases (IBD) are chronic gastrointestinal disorders that are not curable at present. The two main types of IBD are ulcerative colitis (UC) and Crohn’s disease (CD). UC only manifests in the colon as a continuum inflammation that generally starts in the distal colon, going forward through the proximal colon until the cecum, and can lead to ulcerations and bleeding. On the other hand, CD can appear as patched lesions in any part of the gastrointestinal tract, associated with inflammation, stenosis, and/or fistulas. These diseases are characterized by a relapsing behavior, manifested by alternating phases of inactive states in which there is no intestinal inflammation (called remission or quiescence), and active states that present inflammation or any other disease symptoms (an active disease) [1]. In addition, IBD patients often receive multiple treatments and the response to treatment is highly variable depending on the subject, demonstrating the need for a deeper understanding of the disease as well as innovative therapeutics approaches. Growing insights into immunomediated diseases pathophysiology, such as IBD, have led to the advent of targeted therapies, which could selectively interfere with crucial mediators of the inflammatory process [2,3,4,5].

IBD is a multifactorial disease with an unknown etiology. IBD presents defects in the detection and control of the gut microbiota, associated with unbalanced immune reactions, genetic mutations that confer susceptibility to the disease, and complex environmental conditions such as westernized lifestyle [6,7]. Moreover, IBD susceptibility is associated with polymorphisms in host microbial sensory genes such as NOD2 and Toll-like receptor (TLR) 4 [8]. Therefore, currently the most accepted etiopathogenic theory is that IBD is caused by an impairment in the immunological tolerance, resulting in an exacerbated immune reaction against the gut microbiota in genetic-susceptible individuals. Nonetheless, whether the unbalanced immune reactions are the cause or the consequence of the intestinal dysbiosis observed in IBD patients remains to be elucidated.

The human gastrointestinal tract harbors a complex and dynamic community of microbes, namely archaea, eukaryotes, viruses, and predominantly bacteria. This variety of microorganisms constitutes the so-called gut microbiota, while the gut microbiome encompasses these microorganisms along with their genes. The human genome consists of about 23,000 genes, whereas the microbiome encodes over three million genes, which replace many host functions, consequently influencing the host phenotype and health [9].

There is a great relationship between the human host and its gut microbiota. Gut microbiome research has identified three primary functions of the intestinal microbiota: (1) Nutrition and metabolism functions, as a result of the biochemical activity of the microbiota, which include energy recovery in the form of short-chain fatty acids (SCFA), vitamin production, and favorable effects on calcium and iron absorption in the colon; (2) protective functions, preventing the invasion of infectious agents or overgrowth of resident species with pathogenic potential; and (3) trophic effects on intestinal epithelial proliferation and differentiation, affecting neuroendocrine pathways, and on immune system development and modulation [10]. The gut microbiota is a component of the gut barrier, a pivotal complex structure which acts as a frontier between the host and the environment, thus regulating the interaction between the host and bacteria, and modulating nutrient absorption [11]. Consequently, any alteration of the microbiota can lead to a number of gastrointestinal disorders and metabolic diseases. Figure 1 summarizes gut microbiome functions in healthy adults.

Advances in DNA sequencing technologies have significantly contributed to our knowledge of the complexity of this ecosystem. The actual picture indicates that there is no single healthy microbiome since microbiome characteristics are different for each individual. In general terms, microbial balance is needed in order to optimally support metabolic and immune functions as well as prevent disease development. In a healthy gut, pathogenic and symbiotic microbiota coexist without problems. However, any disturbance in that balance leads to dysbiosis, thereby altering normal interactions between microorganisms and the host. As a result, the host may become more susceptible to disease [13]. While the disruption in the equilibrium of the intestinal milieu in IBD is widely accepted, alterations involving biological mechanisms driving dysbiosis remain unknown and it is unclear whether dysbiosis represents a cause or consequence of the disease.

In this review we will focus on how the development of microbiota and the immune system during life contributes to establishing immunological tolerance. In addition, we present IBD as a paradigm of how defects in this process can lead to disease.

## 2. Methods

Data were obtained from articles published in English belonging to journals indexed in PubMed from inception to December 2020. Included search terms were related to (1) immune system maturation, (2) gut microbiota establishment, (3) gut dysbiosis and IBD, and (4) microbiota and modulation of immune system.

## 3. Microbiota and Intestinal Immune System Development

Immune system maturation and microbiota colonization are processes generally considered to take place in parallel after birth. Before birth, the fetus develops an immature immune system, whose innate immune cells (dendritic cells, monocytes, macrophages, natural killer cells, innate lymphoid cells, neutrophils) are generated at different time-points during gestation. These innate cells show low responsiveness to antigens [14] and reduced pro-inflammatory cytokine production [15]. This “muted” immune system is probably necessary to ensuring tolerance to antigens and antibodies transmitted from the mother to the fetus, and also important to facilitating other physiological processes during in utero development. Upon birth, adaptive immune cells (T and B lymphocytes) are mostly naïve, although they also include relevant numbers of regulatory T cells (Treg). Besides, the fetus and neonate’s immune response is skewed toward a Th2 phenotype, reinforcing the tolerogenic behavior of the immune system [14,15]. It is noteworthy that the generation of immune memory needs the contact with foreign antigens; however, since in utero the fetus is only exposed to maternal antigens memory lymphocytes are scarce before birth. Likewise, immunoglobulin M (IgM) is predominant in the neonate because the immunoglobulin class-switch in B cells rarely takes place during gestation [15]. Besides, some secondary lymphoid tissues (such as spleen, lymphoid nodes, and Peyer’s patches) are developed during gestation, whereas others (cryptopatches and isolated lymphoid follicles) depend on the stimulation of the immune system by an early microbial colonization, and therefore are formed after birth [16].

Microbial colonization starts at birth, although it may occur even before, as some bacterial species have been found in the placenta, the umbilical cord, and the amniotic fluid. The composition of this initial microbiota is influenced by perinatal conditions (mode of delivery, type of feeding or antibiotic usage), factors associated with the mother (diet, age and metabolic status), the host genetics and the family’s lifestyle [17].

First colonizers are facultative anaerobes that promote an adequate environment for strict anaerobes. Additionally, the newborn diet, consisting only of breast milk or infant formulas, has great impact on the establishment of the infant’s early microbiota (for the purposes of this review, we will focus on the effects of breast milk). Human breast milk main components are proteins, lipids, oligosaccharides (HMO, human milk oligosaccharides) and immune molecules, as well as some bacterial species (such as *Bifidobacterium*) that might be another important microbial source for the infant’s gut [18,19]. Although the human gut lacks enzymes to digest HMO, members of Bifidobacteria species can use them as an energy source. Therefore, HMO composition influences the selection of microbial species that are nurtured in the infants’ intestine; in addition HMO are also able to inhibit microbial adhesion to and invasion of the gut mucosa, offering an additional protection against infections [20].

Bifidobacteria degrade HMO into monosaccharides and oligosaccharides that support the growth of other microbes, leading to the establishment of the microbial community. Consequently, the microbiota is distributed along the gastrointestinal tract depending on factors such as acidity, oxygen tension, transit time, and nutrient absorption [21]. Thus, microbes are less abundant in the stomach (where acidity compromises their survival) and become progressively more abundant along the gastrointestinal tract until reaching their highest amounts in the colon [22]. The assembly of these microbial communities within the gastrointestinal tract during early life plays a critical role in immune, endocrine, and metabolic host functions, among others [23].

Fermentation of HMO by Bifidobacteria also produces SCFA, which are an important source of energy for enterocytes and are key signaling molecules for the maintenance of gut health and immune tolerance. Additionally, HMO have been suggested to have anti-inflammatory properties, which could be implicated in the development and maturation of the intestinal immune system [20]. Moreover, they are able to modulate intestinal epithelial cell responses, induce apoptosis, and promote a balanced Th1/Th2 cytokine production [19].

Breast milk, as previously mentioned, also contains immune molecules and antimicrobial components, including anti-inflammatory molecules (such as interleukins [IL] 1, 6, 8 and 10; and transforming growth factors), molecules with the potential to mediate B cell growth and differentiation, and modulators of pattern-recognition receptors (soluble TLR2 and 4, soluble CD14) that might help in the initial establishment of beneficial microbiota in the neonate [18]. Moreover, human breast milk contains a large amount of maternal immunoglobulin A (IgA), which is its most abundant immune molecule, together with maternal immunoglobulin G (IgG). Altogether, these molecules will passively protect the breast-fed baby against infections while his/her immune system is maturing [14,18].

Few months after birth, gradual introduction of solid food until weaning is completed is accompanied by a change in gut microbiota composition. Milk-consuming bacteria such as Bifidobacteria species become less abundant, while butyrate producers belonging to the Bacteroidetes and Firmicutes phyla significantly increase. As a consequence, during early infancy gut microbiota composition is less diverse and highly variable; however, around 2–5 years of age the composition, diversity and functional capabilities of gut microbiota resemble those of adults [17].

Early gut colonizers are able to prime mucosal T cells, generating not only immune memory but also promoting the development of T helper cells (Th1, Th2, Th17) and Treg. Upon weaning, the mother’s passive protection fades, and the infant becomes more vulnerable to infections. Then, microbes, environmental antigens and vaccines contribute to a gradual maturation of the immune system during infancy [14], generating immune memory that lasts for decades. All of these interactions also promote immunoglobulin isotype class-switch and the development of the immunoglobulin repertoire, conferring a long-lasting humoral protection mediated by both plasma cells and memory B lymphocytes [14,15].

In adulthood, the microbiome is relatively stable, however is highly variable among individuals and it is subject to perturbation by life events such as diet, medication, exposure to pathogens, age, stress/anxiety, physical activity, tobacco use, or alcohol consumption; then, it declines in old age [21]:

The normal human gut microbiota is primarily composed of two dominant bacterial phyla, Firmicutes and Bacteroidetes, that represent more than 90% of the community, and by other less abundant phyla including Proteobacteria, Actinobacteria, and Verrucomicrobia [24]. Even though there is a common core, composed predominantly of the aforementioned bacterial phyla, the composition and diversity of gut microbiota vary along the gastrointestinal tract, showing a steady increase in the microbial concentration from small numbers in the stomach to very high concentrations in the colon [13,25]. In addition, within the gut, there is also a difference in microbial populations between mucosal surfaces and the lumen [26,27]. Microbes at the mucosal surface are in closer proximity to the intestinal epithelium and may have a greater influence on the immune system, whereas luminal/fecal microbes might be more essential for energy and metabolic interactions [26].

Microbiota composition changes in the elderly (over 65 years old), who present a decrease in anaerobic bacteria such as *Bifidobacterium* spp. and an increase in *Clostridium* and Proteobacteria [21]. Aging is also associated with the process of immunosenescence, which is characterized by an aberrant immune response usually associated with inflammation. This unbalanced immune reaction in the elderly may also impact the relation between host and microbiota, altering microbiota diversity, as well as impairing the development of tolerance to self-antigens, thereby leading to autoimmune disorders [14].

## 4. Gut Dysbiosis in Inflammatory Bowel Disease

### 4.1. Microbial Gut Dysbiosis

Dysbiosis refers to an imbalance in microbial species abundance, which is commonly associated to impaired gut barrier function and inflammatory activity [28]. While some microorganisms are considered essential regulators of the immune system, others can trigger proinflammatory pathways and cause diseases. Major traits of dysbiosis are loss of beneficial microbes, expansion of pathobionts, and loss of microbial diversity [13]. Gut dysbiosis is linked to many diseases, including IBD [29], type 2 diabetes [30], cardiovascular diseases [31], and neuropsychological conditions triggered through the “gut-brain axis” [32]. Yet its role and dynamics in health and disease are poorly understood.

To date, numerous microbiome surveys have been conducted to identify the gut microbiome profile in IBD, especially focusing on the differences between the profile of IBD patients and that of healthy controls. Compelling studies in animal models and humans have provided evidence of persistent imbalance on the gut microbiome in IBD [29,33,34,35,36,37,38,39,40,41,42,43,44,45,46,47,48,49,50,51,52,53,54,55,56,57,58,59,60,61,62,63,64,65,66,67,68,69,70,71,72,73,74,75,76,77,78,79,80,81,82]. However, it remains to be determined whether these changes in the microbiome are the cause of IBD or rather the result of inflammation after IBD onset. Table 1 shows the cardinal features consistently found in IBD dysbiosis.

Current research on microbiome is mainly focused on bacteria, however the gastrointestinal tract is colonized by trillions of microorganisms that include bacteria, archaea, fungi, and viruses. These non-bacterial microbial communities also play a vital role in host health and disease [83,84,85]. It is well established that IBD is affected by several types of microbial species, including fungi, archaea, bacteria, and viruses. Accordingly, deciphering the function and composition of the human gut microbiome in the progression of chronic inflammation in the intestine is crucial to further understand the pathogenic mechanisms of IBD. Table 2 shows the immune pathways that may be affected by microbial taxa altered in IBD.

In the next sections we will briefly summarize the gut dysbiosis associated to each of the four aforementioned microbial communities.

### 4.2. Bacterial Dysbiosis

Bacterial gut microbiome data show that there is a disease-dependent reduction of biodiversity and an imbalanced bacterial composition in the gut of IBD patients compared to healthy controls. The gut dysbiosis profile is also different between IBD patients undergoing different disease activities (flare or remission) [49,53,54,67,100,101,102,103,104,105,106].

The main feature in IBD gut dysbiosis is the decrease in beneficial bacteria. The number of SCFA-producing bacteria such as *Faecalibacterium prausnitzii*, *Roseburia,* or *Eubacterium* is reduced, which may lead to disease, since SCFA strengthens the intestinal barrier and the immune system, thereby contributing to fight pathogens [86,87,88]. In the Bacteroidetes phyla, *Bacteroides fragilis*, a bacterium that can induce Treg growth and expression of cytokines with protective effects against colitis [89,107] has been consistently shown to be decreased in IBD [46,80,108,109,110]. Depletion of *Bifidobacterium*, another beneficial genus in the Actinobacteria phyla exerting important functions in gut homeostasis and health [90,111], has been also reported in the gut of IBD patients [59,112,113,114,115,116,117]. However, controversial results were found within this genus, as other authors also reported its increase in IBD patients compared to controls [47,48,63,100]. A plausible explanation is the effect of disease activity, as *Bifidobacterium* genus is significantly decreased in stool samples of active CD and UC compared to the inactive state [105,118,119]. Nevertheless, further research is needed to elucidate the role of this protective genus in IBD pathogenesis.

In parallel, a significant increase in some pathogens such as Proteobacteria (adherent-invasive *Escherichia coli, Pasteurellaceae*), Firmicutes (*Veillonellaceae* and *Ruminococcus gnavus*), and *Fusobacterium* species has been widely reported. Especially, the increase of the phylum Proteobacteria, which is associated with a proinflammatory state [106] and includes multiple genera considered potentially pathogenic such as *Escherichia*, *Salmonella*, *Yersinia*, *Desulfovibrio*, *Helicobacter,* or *Vibrio*, has been extensively reported in IBD patients [35,43,45,46,47,48,49,50,51,52,79]. The most recurring and contrasted finding is the increase of adherent-invasive *E. coli* in the gut of IBD patients; this infectious agent is able to adhere to and cross the intestinal mucus barrier, invading the gut epithelial layer. Moreover, this species is capable of surviving and replicating in macrophages, leading to TNFα secretion and inflammation [92,120].

### 4.3. Fungal Dysbiosis

Fungi constitute approximately 0.1% of the total microbial community in the gut [24] yet changes in gut mycobiota have been also reported in IBD patients. The innate immune response against molecules in fungal cell walls is complex and incompletely characterized [84,121]. Fungi interact with host immune system via Toll-like receptors (TLR2 and TLR4 predominantly), dectin-1 (CLEC7A), scavenger receptor family (CD5, CD36, and SCARF1), and components of the complement system, which can be activated by fungal cell wall glycoprotein components, such as beta-glucans, chitin, and mannans. Such interaction leads to immune signaling via molecules such as CARD9, IL17, IL22, NF-κB, NFAT, and ITAM containing receptors [84,97]. For example, dectin-1-deficient mice show increased susceptibility to chemically induced colitis, probably due to a disturbed ability to mount effective immune response against the commensal fungal community of the intestinal microbiota [122].

Studies on changes in fungal diversity between IBD patients and controls have shown controversial results. While in some cases fungal diversity was shown to be decreased in UC patients [69,123], in others diversity and richness was reduced in CD [70,72,73,124], and even showed no difference between IBD patients and control groups [68]. An increase in fungal load, especially in *Candida albicans* [68,69,70,71,72,73], is the most solid conclusion across studies.

Literature has shown interkingdom correlations among *Candida tropicalis, E. coli,* and *Serratia marcescens* in CD patients [124]. This network between bacteria and fungi opens a new avenue for research in gut microbiome dysbiosis and may increase our understanding of the underlying mechanisms and role of the gut mycobiota in the development of IBD.

### 4.4. Viral Dysbiosis

The human gut virome includes a diverse collection of viruses directly impacting on human health, including physiological members of the healthy gut microbiota, mostly bacteriophages and eukaryotic viruses [83]. DNA sequencing technology showed that bacteriophages represent the most abundant members of the human gut virome [125]. Viruses can provide bacteria with genes encoding for different functions, and such interkingdom interactions can confer genetic variations to the host microbiome that may contribute to establishing specific phenotypes. This underlines the importance of gut virome in genotype–phenotype studies and suggests a crucial role of viruses in the host [98].

In stool samples, Norman et al. described the enteric virome and its specific alteration in IBD. These researchers showed that *Caudovirales* and *Microviridae* are the most abundant families of bacteriophages in the enteric virome. They revealed that IBD is associated with significant expansion of *Caudovirales* [76]. In gut biopsies, phage populations are increased and significantly different in patients with IBD when compared to controls [74,75].

Pérez-Brocal et al. conducted a linear discriminant analysis effect size (LEfSe) with differential viral discriminant features showing statistical significance for several species in feces from patients with CD compared to healthy controls; especially, increased numbers of overrepresented viruses were observed in feces from patients with CD [126]. They also found that gut virome modification in newly diagnosed IBD patients could be associated with inflammation and linked to bacterial dysbiosis. This observation is in accordance with other investigations in which an inverse correlation was observed between IBD-associated changes in the virome and bacterial microbiome, suggesting a possible model where changes in the gut virome may affect bacterial dysbiosis and/or intestinal inflammation [75,76]. Besides, certain eukaryotic viruses might trigger intestinal inflammation and contribute to IBD pathogenesis [99,127,128].

### 4.5. Archaeal Dysbiosis

Prokaryotes forming the domain of Archaea can also colonize distinct niches in the human body, including the gut. Methane-producing archaea (methanogens) play an important role in digestion, improving polysaccharide fermentation by preventing accumulation of acids, reaction end products, and hydrogen gas [129].

Some studies associated an altered proportion of archaea with IBD. Lecours et al. [130] showed that the abundance of *Methanosphaera stastmanae* in fecal samples was significantly higher in IBD patients than in healthy subjects. Interestingly, only IBD patients developed a significant anti-*Msp. stadtmanae* IgG response, indicating that the composition of archaeal microbiome appears to be an important determinant of the presence or absence of autoimmunity. Another study demonstrated an inverse association between *Methanobrevibacter smithii* load and susceptibility to IBD, which could be extended to IBD patients in remission as *Mbb.smithii* load was found to be markedly higher among healthy subjects in comparison to IBD patients [131]. Controversial results were found by Chehoud et al. [132] who showed no alterations in the archaeal colonization of the gut associated with IBD and found that archaea seemed to be rare in pediatric samples compared to those from adults.

## 5. The “Hygiene Hypothesis”, Dysbiosis and Inflammatory Bowel Disease

The “hygiene hypothesis” suggests that a lack of early childhood exposure to symbiotic microorganisms and helminthic parasites affects immune development increasing the susceptibility to immune-mediated diseases later in life. Weinstock et al. [133] suggested that urbanization and environmental changes toward a more hygienic status diminished the prevalence of helminth colonization in the host leading to a higher incidence of IBD. This hypothesis agrees with findings of Deepshik et al. [134], who demonstrated that helminth infection protects mice deficient in the CD susceptibility gene NOD2 from intestinal alterations by inhibiting colonization by inflammatory *Bacteroides* species. In murine and human studies, they demonstrated that infections with gastrointestinal helminths can protect against IBD by causing immune responses that alter the balance of commensal and pathogenic bacteria in the gut. Another parasites, such as *Blastocystis*, were significantly less frequent in UC patients as compared to healthy controls [135], which reinforces the “hygiene hypothesis.”

Helminths and other parasites are not assessed in microbiome studies as they do not fall in the “microorganisms category,” however its inclusion may improve the understanding of the immune mechanisms underlying IBD pathogenesis.

An interesting aspect of the “hygiene hypothesis” is the fact that the proposed action of symbiotic microorganisms and parasites on the immune system takes place during infancy, which is the critical time-period when a healthy symbiotic relationship is established between the host and the intestinal microbiota. Therefore, it is a matter of debate whether there might be a “window of opportunity” [136] to restore any alteration in the microbial colonization and the subsequent immune imprinting, in order to prevent chronic inflammatory diseases such as IBD (nicely reviewed by Nabhani and Eberl (2020) [136]).

## 6. Conclusions

Correct interplay between gut microbiota and the host is essential for human health. Microbial balance is pivotal for host metabolic and immune functions as well as to prevent disease development. Disturbance in that balance generates dysbiosis making the host susceptible to certain diseases. Gut microbiota stimulates the immune system, and altered composition of this microbiota in early life can lead to an inadequately trained immune system that can overreact to commensal microbes and lead to inflammatory diseases.

Recent research has provided striking findings supporting that the gut microbiome plays an important function in the etiopathogenesis of IBD. Most of the available evidence comes from studies on bacteria, whereas data on the role of fungi, viruses, or archaea are limited. Modifications in specific microbial species, affecting both their diversity and stability, have been identified in IBD.

These microbial alterations of the gut may cause dysregulated mucosal immune responses leading to the onset of IBD, as many of the altered taxa have a direct impact on certain immune pathways, specially favoring a proinflammatory environment. The functional significance of these changes and their pathogenic role remain to be discovered.

The complex interplay between the microbiota, the intestinal mucosa, and the immune system highlights the importance of a comprehensive approach to unravel the mechanisms underlying intestinal dysbiosis.

## Figures and Tables

**Figure 1 ijms-22-03076-f001:**
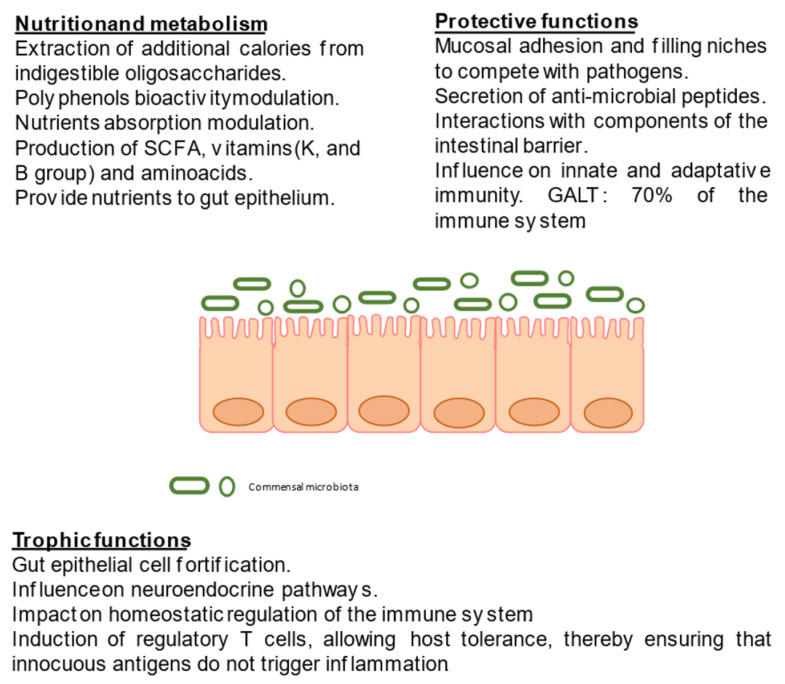
Gut microbiome functions in healthy adults. Adapted from Aziz et al. (2013) [10] and Rowland et al. (2018) [12]. Abbreviations: SCFA, short-chain fatty acids; GALT, gut-associated lymphoid tissue.

**Table 1 ijms-22-03076-t001:** Main features consistently found in inflammatory bowel disease gut microbiome. Summarized from [34,35,36,37,38,39,40,41,42,43,45,46,47,48,49,50,51,52,53,54,56,57,58,59,60,61,62,63,64,65,67,68,69,70,71,72,73,74,75,76,78,79,80,81,82].

Features Consistently Found in Inflammatory Bowel Disease Gut Microbiome
Decreased diversity
Reduced community stability
Decrease in the Firmicutes Phyla and increase of Bacteroidetes
Decrease in *Clostridium XIVA* and *IV* (*Faecalibacterium*, *Lachnospiraceae*, Clostridiumcocoides) and increase in Proteobacteria (Enterobacteriaceae)
Decrease in *Faecalibacterium prausnitzii* and *Roseburia hominis*
Increase in *Ruminococcus gnavus* and adherent-invasive *Escherichia coli*
Increased fungal abundance, mainly *Candida albicans*
Increase of *Caudovirales* and eukaryotic viruses
Bacteriome and virome correlations

**Table 2 ijms-22-03076-t002:** Gut microbiome in inflammatory bowel disease and its associations with the immune system.

Depleted	Immune Association
SCFA producing bacteria (*F. prausnitzii, Roseburia, Eubacterium)*	Produce SCFA playing a major role in modulation of inflammation, regulation of immune responses, maintenance of barrier integrity in the gut, enhanced expansion of Treg population and skew of human dendritic cells to prime IL-10-secreting T cells [86,87,88].
*B. fragilis*	Produces lipid antigens controlling homeostatic iNKT cell proliferation and activation [89].
*Bifidobacterium*	Inhibits intestinal inflammation by acting on Treg cells [90].
*Mbb. smithii*	Weak association with proinflammatory mechanisms [91].
**Enriched**	**Immune association**
*E. coli* (adherent invasive)	Invades intestinal epithelial cells, replicates in macrophages and induces granulomas [92].
*Proteobacteria (* *Salmonella, Yersinia, Desulfovibrio, Helicobacter, Vibrio)*	Associated with a proinflammatory state as revealed by quantification of common proinflammatory interleukins. The inflamed gut appears to provide a favorable environment for expansion of this phyla [93].
*R. gnavus*	Secretes a complex glucorhamnan polysaccharide inducing TNFα secretion by dendritic cells [94].
*Fusobacterium*	Especially *F. nucleatum,* which is a well-recognized proinflammatory bacterium [95] and it may secrete Outer Membrane Vesicles (OMVs) that activate epithelial TLR4 to drive inflammation [96].
*C. albicans*	Interacts with mucosal innate immune cells through the pathways associated with Dectin-1 in macrophages [97].
Bacteriophages *(Caudovirales* and *Microviridae)*	May play a direct role in intestinal physiology or change the bacterial microbiome through predator-prey relationships [76]. Enterobacteria are the main hosts of Microviridae [98].
Eukaryotic viruses	Infect host cells and may increase host susceptibility to IBD by supporting a long-standing immune response through inflammatory mediators, as well as by inducing alterations in the composition of the commensal microbiota [99].
*M. stastmanae*	Leads to substantial release of proinflammatory cytokines in monocyte-derived dendritic cells [91].

Abbreviations: Short chain fatty acids (SCFA), invariant natural killer T (iNKT) cells, Toll-like receptor 4 (TLR4), Inflammatory bowel disease (IBD).

## Data Availability

All data described in the review are included in this published article.

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
