# Peer review of "The Interplay between Immune System and Microbiota in Inflammatory Bowel Disease: A Narrative Review"

_ijms, 2021, doi:10.3390/ijms22063076_

Round 1

Reviewer 1 Report

This paper is based on the role of the gut microbiota in the development of the host immune system and the consequences of gut dysbiosis in intestinal homeostasis.

This paper is well written. I have just a few suggestions.

-  Line 179 to 181, the authors could add a reference.

- The authors were clear that it is to be determined whether the dysbiosis is the cause of IBD or the inflammation in IBD that leads to dysbiosis. However, the title utilized in this paper may suggest that the IBD is the cause of dysbiosis. I would suggest the authors to alter the title.

- The Figure 1 summarize the gut microbiome functions in healthy adults. However, this paper is not only talking about physiologic conditions. I would suggest the authors to create an illustration including health and under IBD condition encompassing all the dysbiosis conditions mentioned in this paper, like bacterial, fungal, viral, and archaeal.

Author Response

Response to Reviewer 1 Comments.

This paper is based on the role of the gut microbiota in the development of the host immune system and the consequences of gut dysbiosis in intestinal homeostasis.

This paper is well written. I have just a few suggestions.

Point 1: Line 179 to 181, the authors could add a reference.

Response 1: A reference [26] has been added in line 196.

Point 2: The Figure 1 summarize the gut microbiome functions in healthy adults. However, this paper is not only talking about physiologic conditions. I would suggest the authors to create an illustration including health and under IBD condition encompassing all the dysbiosis conditions mentioned in this paper, like bacterial, fungal, viral, and archaeal.

Response 2: Thank you for tour comment. Figure 1 is focused on gut microbiome functions in healthy adults. This information has been complemented by table 1, entitled “Main features consistently found in inflammatory bowel disease gut microbiome”, it summarizes the evidence on dysbiosis conditions in, where the main features in IBD gut microbiome have been cited.  We did it in that way to avoid presenting redundant information.

Please note that the references number has changed due to the revisions. 

Thank you very much for your comments and for giving us the opportunity of sending a reviewed version of the manuscript. We have addressed all the concerns following your suggestions as much as possible. We feel that your comments have considerably improved the final version of the manuscript.

Reviewer 2 Report

A very interesting paper examining inflammatory bowel disease microbiome profile; I have some queries:

A material and methods section in my opinion would give this paper major importance; 

Also , a flow chart describing how the studies were included in this paper, would be a great add to the work; alternatively, the suffix "a narrative review" shloud be added to the title.

In the introduction section, a small paragraph describing current innovative  treatments would be a great add; here some works you should consider: doi: 10.1111/dth.12811. and doi: 10.1371/journal.pone.0241575.

Thank You

Author Response

Response to Reviewer 2 Comments

A very interesting paper examining inflammatory bowel disease microbiome profile; I have some queries:

Point 1: A material and methods section in my opinion would give this paper major importance

 Response 1: A methods section has been included. in lines 96-100. We included a descriptive paragraph indicating how we collected the studies.

Point 2: Also, a flow chart describing how the studies were included in this paper, would be a great add to the work; alternatively, the suffix "a narrative review" should be added to the title.

Response 2: We included the suffix "a narrative review" to the title.

Point 3: In the introduction section, a small paragraph describing current innovative treatments would be a great add; here some works you should consider: doi: 10.1111/dth.12811. and doi: 10.1371/journal.pone.0241575.

Response 3: Thank you for your suggestion. We included a new paragraph referring to this issue in lines 38-43.

Please note that due to the revision the reference numbers have changed.

Thank you very much for your comments and for giving us the opportunity of sending a reviewed version of the manuscript. We have addressed all the concerns following your suggestions as much as possible. We feel that your comments have considerably improved the final version of the manuscript. 

Round 2

Reviewer 2 Report

the authors responded to all queries. The paper is publishable.